# Potential Role of the Antidepressants Fluoxetine and Fluvoxamine in the Treatment of COVID-19

**DOI:** 10.3390/ijms23073812

**Published:** 2022-03-30

**Authors:** Mohamed Mahdi, Levente Hermán, János M. Réthelyi, Bálint László Bálint

**Affiliations:** 1Department of Biochemistry and Molecular Biology, Faculty of Medicine, University of Debrecen, Egyetem tér 1, 4032 Debrecen, Hungary; mohamed@med.unideb.hu; 2Infectology Clinic, University of Debrecen Clinical Centre, Bartók Béla út 2-26, 4031 Debrecen, Hungary; 3Department of Psychiatry and Psychotherapy, Semmelweis University, Balassa utca 6, 1083 Budapest, Hungary; herman.levente@med.semmelweis-univ.hu; 4Department of Bioinformatics, Semmelweis University, Tűzoltó utca 7-9, 1094 Budapest, Hungary

**Keywords:** COVID-19, SARS-CoV-2, fluoxetine, fluvoxamine, acid sphingomyelinase, SSRI, sigma-1 receptors, FIASMA, SLC22A3, lysosomotropic agents, clinical studies, drug repurposing

## Abstract

Mapping non-canonical cellular pathways affected by approved medications can accelerate drug repurposing efforts, which are crucial in situations with a global impact such as the COVID-19 pandemic. Fluoxetine and fluvoxamine are well-established and widely-used antidepressive agents that act as serotonin reuptake inhibitors (SSRI-s). Interestingly, these drugs have been reported earlier to act as lysosomotropic agents, inhibitors of acid sphingomyelinase in the lysosomes, and as ligands of sigma-1 receptors, mechanisms that might be used to fight severe outcomes of COVID-19. In certain cases, these drugs were administered for selected COVID-19 patients because of their antidepressive effects, while in other cases, clinical studies were performed to assess the effect of these drugs on treating COVID-19 patients. Clinical studies produced promising data that encourage the further investigation of fluoxetine and fluvoxamine regarding their use in COVID-19. In this review, we summarize experimental data and the results of the performed clinical studies. We also provide an overview of previous knowledge on the tissue distribution of these drugs and by integrating this information with the published experimental results, we highlight the real opportunity of using these drugs in our fight against COVID-19.

## 1. Introduction

The COVID-19 pandemic has spread around the globe in several waves and close to 500 million confirmed cases have been reported, with more than six million deaths as of March 2022 [1]. As vaccinations became available in early 2021, by March 2022, roughly 65% of the world’s population has received at least one dose of vaccine, but 85% of people living in low-income countries have not yet received their first dose [2].

Severe COVID-19 develops in several steps and might require hospitalization or in many cases intensive care. It is of major importance to identify ways that can prevent severe COVID-19 and decrease the burden on the healthcare system in general and intensive care treatment facilities in particular. Several potential medications were tested during the pandemic period based on prior knowledge about the life cycle of the SARS-CoV virus [3]. Drugs known to be lysosomotropic agents were tested, such as azithromycin, chloroquine, hydroxychloroquine, and ivermectin, but were found to be ineffective in preventing severe COVID-19 and results of clinical trials on ivermectin use were also disappointing [4,5,6]. Other drug repurposing efforts were highly successful; such as using steroids to prevent the cytokine storm, budesonide to reduce local inflammation in the lung, and anti-IL-6 inhibitors, such as tocilizumab, to prevent the devastating results of an already initiated cytokine storm [7].

In this review, we systematically present the acquired knowledge about the potential use of fluoxetine and fluvoxamine in preventing the development of severe COVID-19. We discuss the mechanisms, and the clinical and experimental results that underline the potential benefit of these medications in COVID-19 treatment. Moreover, we extensively discuss the previously acquired knowledge about the concentration of fluoxetine in different tissues. It is important to note that among all tissues, the lung tissue is a place of preferential accumulation of fluoxetine reaching concentrations that exceed the concentrations of plasma or even brain tissue. We present molecular data on the transport of fluoxetine into different tissues and refer to their relevance in COVID-19 treatment. Next, we connect these body fluid concentrations with the published experimental data in order to understand the potential clinical relevance of the published experimental results. Based on these, we summarise the major findings of the reported clinical studies and briefly discuss the clinical aspects of including these drugs in a potential COVID-19 treatment protocol in the future.

## 2. Molecular Mechanisms

### 2.1. Molecular Mechanisms of SARS-CoV-2 Replication and the Involvement of Lysosomes in the Replication Process

SARS-CoV-2 is a beta Coronavirus that is a member of the Coronaviridae family, a large 26.0–32.0 kb enveloped RNA virus with a positive-sense single-stranded RNA genome. The life cycle of SARS-CoV-2 is truly complex with steps that are regulated in both space and time, and, indeed, we currently have adequate information on its structural biology and pathogenesis exemplified by extensive reviews [8,9]; therefore, we are only opting to highlight important steps that are crucial for the cohesion of this manuscript. Briefly, following attachment to host target cell receptors, among which angiotensin-converting enzyme 2 (ACE 2) is considered primary, engagement of the ACE 2 receptors results in a conformational change to the Spike (S) protein, followed by subcleavage by furin and target-cell proteases, such as TMPRSS2 and cathepsin L [10,11].

Entry into the cytoplasm and release of the viral genome is followed by the expression of viral polyproteins that are proteolytically processed into four structural and sixteen non-structural proteins, mediated by the viral papain-like protease (PLpro), the chymotrypsin-like protease (3CLpro) that is also referred to as Main protease (Mpro) [12]. Viral polyproteins are subsequently processed by the viral proteases into 16 non-structural proteins, which are crucial to viral replication and transcription [13]. Additionally, a number of subgenomic mRNAs, nested negative sense RNAs resulting from discontinued transcription of genomic RNA, were found to code for accessory proteins among others, which were linked to host cellular immune responses [14,15,16]. Akin to other Beta Coronaviruses, it is thought that SARS-CoV-2 structural proteins, in addition to viral RNA and other non-structural and accessory proteins, form replication complexes, assembling at sites close to the endoplasmic reticulum (ER) and Golgi compartments characterized by membrane tubules and double-membrane vesicles, possibly derived from the ER [17,18]. Thereafter, viral genomic RNA, along with the nucleocapsid, is thought to translocate to budding sites where other structural glycoproteins are located, followed by assembly and release, utilizing ER trafficking and lysosomal egress [12,19]. It is now apparently clear that the replication cycle of SARS-CoV-2, similarly to other Coronaviruses, heavily relies on the ER, hijacking ER stress responses to facilitate protein translation [20].

### 2.2. Molecular Mechanisms through Which Fluoxetine and Fluvoxamine Might Prevent the Development of Severe COVID-19

#### 2.2.1. Binding to Sigma-1 Receptors

Fluvoxamine was hypothesized to exert its antiviral effects in the context of COVID-19 through different mechanisms. Of the greatest significance in this process is its interaction with the sigma-1 receptors (S1Rs). S1R is a multifunctional chaperone protein located within the ER that mediates signalling cross-talks in the ER-mitochondria and ER-nucleus context. As a chaperone, it facilitates proper folding of newly synthesized proteins, in addition to preventing the accumulation of misfolded proteins; therefore, playing a major role in cell survival and modulating ER Ca^2+^ influx into the mitochondria during cellular stress [21]. Under stressful conditions, i.e., viral infection, the overexpression of S1Rs plays a protective role, reducing ER and oxidative stress, counteracting pro-apoptotic signals through the induction of a wide range of mechanisms that promote cell survival [22].

Given their pivotal role in mitigating cellular stress during a viral infection, S1Rs ligands are frequently explored as potential drugs against viral infections and are currently being studied against SARS-CoV-2. For example, in the context of HCV infection, S1Rs were found to colocalize with non-structural proteins associated with the viral replicase complex [23], and more recently, the nsp6 of SARS-CoV-2 was found to directly interact with S1R, highjacking cellular translation machinery, thus, favouring expression of viral proteins. Molecules targeting sigma receptors such as antipsychotic drugs (haloperidol), and antihistamines (clemastine and cloperastine), all of which are sigma receptor ligands, were found to also exert anti-SARS-CoV-2 activity [24]. Fluoxetine, among other antidepressant drugs, was found to perturb the replication of HCV in an unbiased screening cell culture assay [25]. Additionally, the unfolded protein response (UPR) and autophagy are also ER stress responses that are exploited by SARS-CoV-2, and the S protein in β-Coronaviruses was found to modulate UPR, favouring viral protein synthesis and replication [26,27]. Through its modulatory effect on the ER stress response, fluvoxamine may therefore hinder the replication of SARS-CoV-2 [28].

#### 2.2.2. Lysosomal Membrane Composition as the Potential Site of Action of Fluoxetine and Fluvoxamine in COVID-19

Decreasing the acidity of endosomal pH adversely affects endosomal function and trafficking and inhibits the activity of the endosomal proteases [29]. Modulation of endolysosomal pH could potentially impair the formation of viral replication complexes, in addition to impeding viral trafficking and budding [30]. Both fluoxetine and fluvoxamine, among other SSRIs, are considered to be lysosomotropic agents, and hence, it is plausible that the lysosomotropic and endolysosomal pH-modulating effects of these molecules may indeed show potential beneficial effects in the context of COVID-19.

The role of acid sphingomyelinase (ASM) in augmenting viral infection has long been known and described. Covering a wide range of cellular functions, from cytoskeletal reorganization to proliferation, response to stress, signalling, and induction of apoptosis, acid sphingomyelinase catalyses the hydrolysis of sphingomyelin to phosphorylcholine and ceramide. Being a predominantly lysosomal enzyme, proteolytic activation of the ASM can result in its translocation to the cellular membrane and the generation of ceramide at the extracellular cell surface [31,32]. Ceramide is an integral part of the cellular membrane, modulating its biophysical properties, and is composed of a sphingosine backbone that can be post-translationally modified, yielding complex molecules, including glycosphingolipids (GSL). Although there has been documented evidence of the involvement of ceramide and ceramide-based molecules in mediating several viral infections, such as Rotavirus, HIV, and Influenza A, it is unclear whether ceramide or one of its derivatives are involved in mediating the attachment or internalization of virion particles [33,34,35]. In regards to SARS-CoV-2, the main receptor (ACE 2) was found to locate in lipid rafts [36] and the effect of the ASM inhibition, namely the change in the ceramide content of the lipid rafts, was found to inhibit viral infection of the cells in in vitro studies [37]. SSRIs such as fluoxetine or fluvoxamine were found to accumulate in lysosomes and disturb and attenuate the activity of ASM [38]. Indeed, fluoxetine treatment, through this proposed mechanism, efficiently inhibited the infection of Vero-E6 cell lines with SARS-CoV-2 and vesicular stomatitis virus pseudo viral particles enveloped with SARS-CoV-2 S protein [37,39].

#### 2.2.3. Anti-Inflammatory Effect of SSRI-s

Some of the potentially relevant SSRI antidepressants may also exert their protective effect against severe COVID-19 through the down-regulation of the inflammatory response induced by a viral infection, independently of their action on the S1Rs. The possible mechanism of action proposed was through modulating the immune activity of the macrophages [40], while depression itself is considered by some to be an inflammatory disease [41]. The role of macrophages in the anti-inflammatory activity of antidepressant drugs has been reviewed thoroughly and recently by Nazimek et al. [42].

In severe COVID-19, a state of platelet dysregulation is observed, marked by increased activation, reactivity, and aggregation [43,44]. Both fluoxetine and fluvoxamine block the re-uptake of serotonin from plasma in platelets via the sodium-dependent serotonin transporter (SERT), and by limiting the uptake of serotonin, SSRIs interfere with platelet activation and aggregation, therefore, potentially increasing bleeding time and reducing neutrophil recruitment and inflammation [45]. It is therefore conceivable that SSRIs may be of some benefit in advanced COVID-19, counteracting the hypercoagulable state of platelets observed. The topic is still controversial [46], but based on clinical studies, fluoxetine appears to prevent platelet activation and is more likely to prevent thrombotic events [47].

In an in-vivo multiple sclerosis (MS) rat model, fluvoxamine treatment resulted in a significant increase in the viability and proliferation of neural stem cells, and treatment with physiological concentrations attenuated the severity of encephalomyelitis, manifested by a decrease in serum levels of IFN-γ, and an increase in IL-4, pro-, and anti-inflammatory cytokines, respectively [48].

## 3. Clinical Evidence for the Potential Benefits of Using Fluoxetine and Fluvoxamine in COVID-19 Therapy

There are multiple studies available from the past two years that assess the effectiveness of different SSRI medications for patients with COVID-19 infection. Clinical trials mainly focused on fluoxetine and fluvoxamine, while observational studies investigated the possible beneficial effects of other antidepressants as well.

### 3.1. The First Clinical Trial

The first clinical trial was conducted by Lenze et al. in St Louis, USA, during the very early phase of the pandemic in the summer of 2020 [49]. It used a double-blind, randomized, placebo-controlled design, patients were non-hospitalized adults in a community setting. Symptom onset was within 7 days, and patients with low saturation levels (<92%) and other severe underlying medical conditions (COPD, heart failure, and immunocompromised status, etc.) were excluded from the study. Participants either received 3 × 100 mg of fluvoxamine (if tolerated) or a placebo for 15 days, and they did not receive specific antiviral medication as part of the protocol. The primary endpoint was clinical deterioration, which was defined as a low saturation level (<92%) or dyspnoea. Results showed that no patient receiving fluvoxamine reached the primary endpoint, whereas 6 of 72 on placebo had either shortness of breath or low oxygen levels. Only one of them needed ventilator support and no participants died. There were several limitations to this study (including a low number of participants, a small number of endpoints, and no long-term follow-up, etc.); results, however, were encouraging, and it appeared that fluvoxamine could be used in the early phase of the infection to prevent the deterioration of symptoms.

### 3.2. The First Open-Label, Real-World, Prospective Cohort Study

Based on the initial promising results of the above-mentioned study, Seftel et al. launched an open-label, real-world, prospective cohort study, in which fluvoxamine (2 × 50 mg for 14 days) was offered as a therapeutic option for patients living in an occupational setting, during a mass outbreak in November-December of 2020 [50]. A total of 65 patients accepted fluvoxamine, whereas 48 chose only observation, with no specific treatment as an option. There were no major differences regarding basic demographics between the two groups, however, only 38% of those who received fluvoxamine were asymptomatic, compared to 58% in the observation group. A total of 30% of the patients had one or more chronic medical conditions, and the median age was 42. Results heavily favoured fluvoxamine, as no one on this medication needed hospitalization, contrary to 12.5% (6 of 48) in the observation group (*p* = 0.005). Of the patients, two required intensive care treatment and one of them died. Although initial symptoms were more prevalent in the fluvoxamine group, at day 14 the participants were symptomless, whereas 60% (29 of 48) of the patients in the observation groups still had ongoing symptoms (*p* < 0.001), such as anxiety, memory problems, fatigue, and headaches, etc. As a small-scale, open-label, non-randomized trial, it has its obvious weaknesses; however, it provides valuable information about a real-world setting on how fluvoxamine could be used in occupational settings.

### 3.3. The First Double-Blind, Placebo-Controlled Study

The largest scale, randomized, double-blind, placebo-controlled study was conducted by Gilmar Reis et al. in Brazil [51].

TOGETHER is a multicentre clinical trial in which high-risk outpatients received either fluvoxamine (2 × 100 mg) or placebo for ten days, while other medications (hydroxychloroquine, lopinavir-ritonavir, metformin, ivermectin, doxazosin, and interferon lambda) were also tested during this time, therefore, patients were randomized into other treatment arms as well. Only patients with early symptoms (<7 days) and who had at least one other major medical condition (diabetes, cardiovascular disease, and asthma, etc.), and were thus characterized as belonging to a high-risk group, could participate in the trial. The primary outcome was hospitalization or observation in Emergency Care, which lasted longer than 6 h (without waiting times). Secondary outcomes were—among others—viral clearance, time to clinical improvement, number of days with respiratory symptoms, and mortality. Follow-up was 28 days long. Altogether, 741 patients received fluvoxamine and 756 placebo, and no major differences were present between the groups regarding demography and comorbidities. A total of 11% (79 of 741) of participants in the fluvoxamine group and 16% (119 of 756) of the placebo group had a primary outcome event, most of which were hospitalizations. Per-protocol analysis showed a larger beneficial effect in the fluvoxamine groups, therefore, the Data Safety Monitoring Committee decided that randomization of patients had to be stopped, given the superiority of the active arm compared to placebo. No other investigated active compound proved to be effective in this trial. That trial showed the first strong piece of evidence that supported the use of fluvoxamine in the early stages of COVID-19, although it needs to be taken into consideration that the primary endpoint (the number of hospitalizations and longer observations in ER settings) is determined by multiple factors including anxiety. Fluvoxamine, as an excellent antidepressant, can act as an anxiolytic drug after only one week of treatment, which can cast a bias on this study design.

### 3.4. Observational and Retrospective Studies

#### 3.4.1. A Hungarian, Retrospective, Case-Control Study

Németh et al., at Uzsoki Street Hospital, Budapest, investigated whether fluoxetine improves clinical outcome among hospitalized patients with severe COVID-19 [52].

The rationale behind this approach was that fluoxetine—as a potent antidepressant—can decrease anxiety and improve a patient’s mood during the sometimes lengthy hospital stay, and therefore, would improve the general outcome. Medical records of patients treated between 17 March and 22 April 2021 were analysed retrospectively. Indication of fluoxetine was up to the physician’s discretion, however, as positive experiences with the medications arose, it became part of the protocol, therefore, it was more widely used. Altogether, 110 patients received fluoxetine (20 mg/day) and 159 received treatment as usual (TAU), which included—based on the physician’s decision—favipiravir, remdesivir, and baricitimib. Overall mortality was significantly lower in the fluoxetine group (15/110 vs. 49/159, *p* = 0.002), whereas there were no significant differences between other treatment groups. It should also be noted that although fluoxetine was often co-administered with remdesivir, the effect of fluoxetine seems to be independent of it.

This study has serious limitations, which derive from the retrospective manner used and the fact that patients were not randomized into different subgroups. However, in contrast to the previously mentioned trials, most patients were not in the early phases of their infection when they were given fluoxetine, which suggests that SSRI medication can be effective in later stages of COVID-19 as well.

It should also be noted that antidepressants, regardless of COVID-19 effects, may improve outcomes for patients who are treated in intensive care settings or who are hospitalized over a long duration, because depression itself increases the mortality of this patient group [53].It has been reported that SSRIs can have a neuroprotective effect during COVID-19 [54]. Results of this study are therefore probably partially related to the antidepressant effect of fluoxetine and are not necessarily linked to the COVID-19 pathology itself. Overall, in a clinical context, the improvement of depression and anxiety should be considered an important confounder in relation to somatic improvement, because psychiatric improvement has been shown to have a direct effect on somatic endpoints [55,56]. Therefore, changes in depressive and anxiety symptoms should also be screened in future studies assessing the effectiveness of antidepressants in COVID-19 patients, making the control of this cofactor possible.

#### 3.4.2. The Observational Multicentre Retrospective Cohort Study from the Paris Region

This study was conducted by Hoertel et al. in 2020 based on the Health Data Warehouse “Assistance Publique-Hôpitaux de Paris (AP-HP)” [57]. The study examined the possible beneficial effects of various antidepressants (AD). Only data of hospitalized patients were analysed, which suggests that most of them had moderate or severe symptoms. Patients who received an antidepressant within 48h of hospitalization (N = 345) were compared to the control group (N = 6885), furthermore, the AD group was divided into SSRI (N = 195) and non-SSRI (N = 150) subgroups during the analytic process. The primary endpoints were intubation and death. There were no significant differences in patient characteristics between the matched analysed samples. Results showed that all ADs were associated with a significantly reduced risk of intubation or death among hospitalized patients. SSRIs, such as escitalopram, fluoxetine, paroxetine, and an SNRI, venlafaxine, showed the strongest association with better outcomes, but sample sizes were low in the case of some medications (only one patient received fluvoxamine). Overall, the study suggests that a broad spectrum of ADs can improve clinical outcomes, perhaps with different pathomechanisms, moreover it shows that—as it was seen in the previously mentioned publication—SSRIs can be effective not only in the early but also the later stages of the disease.

#### 3.4.3. The San Francisco Region Health Records Analysis

Oskotsky et al. analysed the health records of 490,373 patients in the USA’s San Francisco region who were diagnosed with COVID-19 and had either been hospitalized, observed in an emergency department, or had received urgent care in the community [58].

Similarly to the previous study, it used a retrospective cohort design, whereby patients were matched by demographic characteristics and major comorbidities. A total of 3401 adults received SSRIs at the time of confirmed COVID-19 infection, of whom 481 patients received either fluoxetine or fluvoxamine. The primary endpoint was death. There was a significantly lower death rate in the SSRI group compared to matched controls (14.6% vs. 16.6%), furthermore, the association was even more robust among patients on fluoxetine and fluvoxamine (10.0% vs. 13.3%); however, mortality among patients who received another SSRI did not differ significantly from the control group (15.4% vs. 17.0%). These results suggest that fluoxetine and fluvoxamine may have a superior effect compared to other SSRIs, however, we should keep in mind that only 10 patients received fluvoxamine, which barely has statistical relevance in this kind of setting.

Even though all available studies support the theory that SSRIs, such as fluoxetine and fluvoxamine, could be effective against the development of severe outcomes of COVID-19 infection, very few randomized, placebo-controlled trials have been launched to verify this. One of these ongoing trials is currently recruiting patients with moderate to severe COVID-19 pneumonia in Hungary at multiple sites under the title “Fluvoxamine Administration in Moderate SARS-CoV-2 (COVID-19) Infected Patients”. A total of 100 participants are planned to be enrolled, all of whom are hospitalized due to COVID-19 symptoms. Contrary to the above-mentioned clinical trials, fluvoxamine is administered for a much longer period (74 days) in this setting. The primary outcome measure is the time to clinical recovery after treatment. It will be interesting to know whether the initial encouraging results seen in the “TOGETHER” study can be replicated amongst hospitalized patients in a later stage of the pandemic, when “treatment as usual” is more developed and effective than during earlier stages of the COVID-19 era.

Table 1 summarizes the published clinical studies.

## 4. Experimental Data That Support the Concept That Fluoxetine Might Be Useful in Treating COVID-19

Several specific experimental investigations have been performed during the COVID-19 pandemic that underlie the possibility that fluoxetine might be beneficial in the fight against the development of severe outcomes of COVID-19 infection.

### 4.1. Experimental Data from the Ursula Rensher Group

In a paper accepted in September 2020, before vaccines became available, the group of Ursula Rensher from Münster (Germany) reported a series of experiments in which they had investigated molecules that interfered with cholesterol accumulation in late endosomes [39].

The model cell lines were Vero E6 cells and polarized bronchial Calu-3 cells. The line of Vero cells was established in 1962 from the kidneys of a normal African green monkey. They are non-tumorigenic cells, widely used in vaccine production in standardized conditions [59]. VeroE6 cells are widely used in experiments that study the pathomechanisms of SARS-CoV-2 infection as they can be easily infected with the virus and the released virus particles from these cells can be easily and precisely quantitated by real-time QPCR measurements [60]. While Vero E6 cells are excellent for assessing viral replication, the Calu-3 cells established in 1975 from a pleural effusion of lung adenocarcinoma [61] are good models for epithelial lung cell modelling.

The authors used fluoxetine, imipramine, amiodarone, and the NPC1 inhibitor U18666A. NPC1 is an intracellular cholesterol transport protein that transports low-density lipoproteins into the late endosomes, mutation of NPC1 can cause Niemann Pick type C disease. As they could show that fluoxetine was active against viral replication for influenza virus strains (EC50 = 1 µM and EC90 = 5–6 µM), they tested its effect on SARS-CoV-2 cells and found that EC90 for SARS-CoV-2 was in the range of 2–4 µM depending on the used cell lines. U18666A could reduce viral titers by 99% at the concentration of 10 µM-s. Imipramine and amiodarone were also effective on viral replication in different cell lines.

The authors investigated cholesterol accumulation in late endosomes using microscopic methods. As a positive control, they used U18666A treated cells. They could show a significant accumulation of cholesterol at a very high dose of fluoxetine of 20 µM-s, but at a lower dose of 5 µM-s the results were not significant. The authors tested the changes in the pH of the endosomes by microscopic methods. In this assay, both 5 µM and 20 µM-s of fluoxetine produced significant changes in lysosomal pH, comparable to those caused by 2–10 µM-s of U18666A.

Finally, they compared the infectivity of Vero E6 cells pre-treated with 5 µM fluoxetine or 10 µM of U18666A. Both pre-treatments produced a significant reduction of infectivity, as assessed by intracellular nucleocapsid detection with microscopic methods.

In their next study, accepted for publication in February 2021, at the time when global vaccination campaigns had just been launched, the same group published a manuscript where they argued that remdesivir, itraconazole, and fluoxetine might have synergistic effects on SARS-CoV-2 infection in vitro [62]. They used the same cell lines as in their previous report. The concentration of fluoxetine used in their study was between 0.5 µM and 2.5 µM. At the highest dose of 2.5 µM fluoxetine concentration, without remdesivir co-treatment, the virus titer reduction was negligible. Although they could show synergies for both the combination of remdesivir with Itraconazole and the combination of remdesivir with fluoxetine, a potential limitation of their experimental work was that remdesivir had a very short half-life in serum. In mice experiments, remdesivir was detectable in serum only half an hour after dosing. It is to be noted that the potentially active metabolites of remdesivir could be detected in serum up to 24 h post-dosing [63]. At the same time, an observation that might underlie the potential synergy between remdesivir and fluoxetine in the lung is, that in the same investigation, the concentration of remdesivir in lung tissue could be detected much longer, up to 24 h with the highest concentration at two hours. The reported concentration of remdesivir in the lung tissue was 0.35 µM. At a similar concentration of remdesivir (0.25 µM) in the Renscher study, 2.5 µM of fluoxetine produced only a modest 10% reduction in the virus titer.

In September 2021 the Reschler group published a study [64] wherein they investigated the synergy of fluoxetine with the GS-441524 nucleoside analogue. GS-441524 is the main plasma metabolite of remdesivir with a plasma half-life of 24 h. Human data on GS-441524 are scarce, the current applications are mainly experimental and veterinary [65]. In this paper, the Rescher group investigated the synergy between various doses of fluoxetine and GS-441524. The same Vero E6 and Calu-3 cells were used. Similar synergies were observed as for remdesivir. The highest used concentration was 2.5 µM of fluoxetine, which had marginal inhibitory effects as seen in their previous report. A combination of 2.5 µM fluoxetine with 1 µM of GS-441524 produced a 99.9% inhibition of virus production on polarized Calu-3 cells.

### 4.2. Experimental Data from the Jochen Bodem Group

In March 2021, the group of Jochen Bodem published a short but important paper [66]. Their results confirmed the findings of the Rescher group, namely that fluoxetine dramatically reduced the viral replication of SARS-CoV-2 in the Vero E6 cell line that originates from the kidneys of a normal African green monkey. Similarly to the results of the Rensher group, in the experiments of the Bodem group, the concentration of 2.5 µM of fluoxetine resulted in the reduction of virus titer by one order of magnitude, approx. 90% inhibition, moreover, 5 µM of fluoxetine had a dramatic effect of more than three orders of magnitude reduction of the viral titer. At these concentrations of fluoxetine, no significant inhibition of cell growth was seen in Vero6 cells. Escitalopram or Paroxetine had marginal effects. The results were quantified by virus-specific QPCR and confirmed by microscopic staining. Fluoxetine did not affect other tested viruses such as RSV, Rabies, HSV-1, and HHV8.

The same paper by Bodem [66] reports a very important experiment assessing the effect of fluoxetine on viral replication in normal human lung tissue preparations. Human, disease-free lung tissue slices of 300 µm width with intact peripheral airways were prepared and cultivated from samples originating from lobe resection due to cancer. The tissue slices were treated with 5 µM of fluoxetine and then infected with SARS-CoV-2. After 3 days, supernatants were harvested, and infectivity was assessed in Vero E6 cells. The resulting virus titers were quantified by QPCR. Fluoxetine treatment of the lung slices at 5 µM-s resulted in a more than two orders of magnitude reduction of viral output in the developed assay, which corresponds to more than 99% of inhibition. These fluoxetine concentrations are in line with the concentrations measured in postmortem human lung samples as described later in point 5.2 of this review.

### 4.3. Enantiomer Indifference of the Antiviral effect of Fluoxetine

A third very important observation is reported in the same paper of Bodem [66] that addresses the stereoselectivity of fluoxetine effects. The currently used fluoxetine is the racemic mixture of both S and R enantiomers. In their report, the authors investigated the viral replication inhibitory effect of both the racemic mixture and the two enantiomers separately. In their experiments they found that there was no difference in the inhibitory effects on the virus between the two enantiomers. This observation is of extreme importance if we take into account our previous knowledge about the specifically psychiatric effects of the two fluoxetine enantiomers, which was accumulated in the 1990s. Although the enantiomers of fluoxetine were not studied extensively, the enantiomers of norfluoxetine, the metabolic product generated by demethylation in the liver and which are also active serotonin reuptake inhibitors, have been well investigated. The topic was examined in several studies assessing rat brains and complemented with studies on human platelets [67,68]. These investigations showed that the S enantiomer of norfluoxetine was over 20 fold more potent than the R enantiomer regarding the SSRI effect of the enantiomers. Interestingly, the less effective R enantiomer, if administered orally at 80 or 120 mg/day, resulted in QT elongations on ECG measurements. These QT elongations were statistically significant, underlying the possibility that the cardiac effects of fluoxetine observed in other studies [69] might involve other mechanisms beyond SSRI activity. It is to be noted that the QT elongation effect of fluoxetine, although reported, is still less pronounced compared to the similar effect of citalopram [70], where these cardiac side effects led to the development of escitalopram, which contains the S enantiomer of citalopram to reduce potential side effects. The observations on the similar inhibitory effect on SARS-CoV-2 replication of both S and R fluoxetine enantiomers, together with the 20 fold higher SSRI effect of the S enantiomer, raise the possibility of the development of an antiviral formulation based on the R enantiomer that might have fewer CNS effects. As this is the first report on the enantiomer indifference of fluoxetine antiviral effects, further studies are needed to confirm these observations.

## 5. Pharmacokinetics of Fluoxetine and Fluvoxamine

### 5.1. Pharmacokinetic Data

In order to assess the opportunity for using fluoxetine or fluvoxamine in blunting the severity of the effects of COVID-19, we need to investigate the pharmacological context of their administration. The two molecules present a quite similar pharmacokinetic profile. As the published literature focuses more on fluoxetine, in our review we will first present the data available about fluoxetine and at the end of this sub-chapter we will provide an overview of the major differences regarding fluvoxamine.

Fluoxetine was introduced into clinical practice in 1987 and since then has been widely used in different concentrations and for a relatively wide spectrum of indications in the field of psychiatric perturbances or altered states [71,72].

A particularly beneficial feature of fluoxetine is that several physiological or pathological states that might need an adjustment of the therapeutic dose do not influence its tissue distribution and local concentrations. Fluoxetine is well absorbed, has a significant first-pass hepatic metabolism and low lipid solubility, and as a result, age, obesity, or renal failure do not dramatically affect fluoxetine pharmacokinetics [73,74]. At the same time, in case of liver impairment, e.g., in alcoholic liver cirrhosis, the pharmacokinetics changed significantly, resulting in a more than two-fold increase in the half-life of the molecule [75].

Another great practical feature of fluoxetine is that its bioavailability is not affected by food. A single dose of oral administration will result in a peak plasma level after 6–8 h, with a maximal CNS efficacy as assessed by EEG, detected after 8–10 h post oral administration. In the liver, a demethylation step occurs that generates Norfluoxetine, which is also active as an inhibitor of serotonin reuptake. The significant first-pass metabolism in the liver that generates norfluoxetine can lead to a single dose of oral fluoxetine of 40 mg-s in different persons achieving different maximal plasma concentrations [73,74] ranging from 15 µg/L to 55 µg/L, which corresponds to 50–150 nanomolar (nM) plasma concentrations. If 60 mgs were administered daily, the steady-state plasma level was achieved relatively late, after 30 days, and 80% was excreted in the urine, while 15% in faeces (with 5% of the radiocarbon used for tracing not found in either urine or faeces). Initially, the recommended dose was 80 mg/day, but later 20 mg/day was shown to be more effective. Moreover, in the elderly age group, 3 times 20 mg weekly was shown to be equally effective.

The pharmacokinetics of fluvoxamine have been well described and summarized in several reviews [72,76]. Fluvoxamine is well and rapidly absorbed after oral administration, reaching a maximum plasma concentration level somewhat earlier than fluoxetine. While the range of t_max_ for fluoxetine is considered to be 6–8 h, for fluvoxamine these values are between 2 and 8 h. After oral administration, 96% of fluvoxamine is absorbed, which is a relatively higher fraction compared to fluoxetine, where 80% is absorbed. Both drugs have a strong first-pass hepatic metabolism. As for excretion, 4% of fluvoxamine and its metabolites are excreted with urine while only 80% of fluoxetine and its metabolites are eliminated through the kidneys, and the rest is eliminated with the faeces. Pharmacokinetics of fluvoxamine are not significantly affected by the status of the kidney, but hepatic cirrhosis does affect the elimination of both drugs. Age does not seem to affect the metabolism of any of these two molecules significantly, so long as therapeutic doses are followed.

### 5.2. Oral Dose versus Body Fluid Concentrations

In order to be able to compare the results of the performed clinical studies and reported experimental data, we need to compare the achieved tissue concentrations and the concentrations used in the different experiments. For performing such a comparison, we need to address the issue that cellular experiments report their concentrations in micromoles (µM), while clinical dosage and tissue concentrations are usually expressed in micrograms (µg) and microgram per gram or microgram per millilitre. The molecular weight of fluoxetine is 309.33 g/mol. The currently used typical dose of 20 mgs of fluoxetine is equivalent to 64.65 nanomoles. Tissue distribution is little affected by the amount of adipose tissue in the patient. Peak plasma concentrations after a single dose of 30–40 mg are in the range of 15–55 µg/L corresponding to 48–177 nano Mols. As for tissue distributions, interestingly, the highest concentration of the drug was found in the lung and liver in early measurements performed in dogs and confirmed in human samples as well [71]. In brain tissue, the levels of fluoxetine seem to be two-fold higher (average 2.6 fold higher in some cases 4.9 fold higher) compared to plasma levels, with significant patient-to-patient variability [77,78].

In a post-mortem assessment of body fluids collected from deceased pilots after aviation fatalities [79], the concentration of fluoxetine from all investigated tissues was the highest in lung tissue samples, exceeding 60 fold the concentration in blood with significant person-to-person variability. Similar values were found for the degradation product norfluoxetine, in which case the concentration in the lung was 59 times higher in the lung tissue compared to blood concentration. The highest fluoxetine concentration was 51.9 µg/g in the lung tissue, with a mean of 19.6 µg/g, these concentrations are in the range of 50 µM concentration, with the lowest measured concentration of 1.56 µg/g corresponding to 5 µM. The measurements from post-mortem lung specimens resulted in drug levels in the same range, or even higher than the concentrations of 2–10 µM-s used in the published cell assays that were investigating the effect of fluoxetine on SARS-CoV-2 virus replication in various cell lines (see details in chapter 4 of this review). The concentrations in plasma in pharmacologic studies were in the range of 100 nM. The postmortem data show 60-fold enrichment in lung tissues, relative to blood, and based on this, we can conclude that the lung concentration is very likely to be in the range of 5 µM, which is similar to the effective concentration reported in cellular assays.

Some lung-specific side effect reports in the early 1990s underlie the possibility that fluoxetine might specifically target the lung. Pulmonary or systemic phospholipidosis caused by fluoxetine was documented in animals and reported in humans [80], while direct fluoxetine-induced lung damage without a known cause was reported several times in early case reports [81,82]. These observations underline the enrichment of fluoxetine and its derivatives in lung tissues that might be of extreme importance in preventing severe effects of SARS-CoV-2 infection.

### 5.3. Tissue Distribution of Fluoxetine Drug Transporters

In order to explain the surprisingly high concentration in the lung and the specific effects reported earlier, we investigated what is known in the literature regarding the transporters responsible for the transport of fluoxetine into different body compartments. Fluoxetine is transported by the protein Organic Cation Transporter 3 (OCT-3), which has the canonical name of SLC22A3. A review on various aspects of organic cation transporters and their involvement in the transport of various drugs in different tissues was published recently [83]. Based on the public human tissue-specific gene expression database of the Human Proteome Atlas, on a single-cell level [84] the highest expression of the gene is seen on alveolar cells type 2, followed by hepatocytes and pancreatic endocrine cells, which for hepatocytes, is in concordance with the strong first-pass hepatic metabolism. Regarding the lung, this might explain the high tissue concentration reported earlier. Interestingly, an observation underlying the transport in pancreatic cells is that fluoxetine-induced beta-cell dysfunction was also reported in cellular systems [85,86] in animal studies [87], and a systematic review performed based on Danish patient registries reported a significantly increased risk of pancreatitis in first-time users of SSRIs [88].

### 5.4. The Potential Target Cells in the Alveolae Responsible for the Protective Effect of Fluoxetine and Fluvoxamine in SARS-CoV-2 Infections

As reported during the first waves of the pandemic, the most severe outcome of COVID-19 disease was the progressive patient deterioration approximately a week after the onset of symptoms with a decrease in oxygen saturation levels and progressive decompensation of the respiratory system.

In order to understand the relationship between the two SSRI molecules investigated and the progression of COVID-19 disease, we will focus on the cellular distribution on the alveolar level of key molecules involved in viral infection and SSRI activity.

Based on the public human tissue-specific gene expression database of the Human Proteome Atlas, on the single-cell level [84], in the lung non-vascular cells, the highest expression of ACE2, the most likely cellular entry point of the SARS-CoV-2 virus, is expressed only on alveolar type 2 cells and the expression level is marginal [89].

Interestingly, in this single-cell-level assessment of RNA levels from the different types of cells present in the alveolae, the highest expression level of SLC22A3 was present, on the type 2 alveolar epithelial cells, while the other cells of the alveolae, namely the alveolar epithelial cells type I, macrophages, club cells, fibroblasts, ciliated cells, and other immune cells of the alveolae had a significantly lower or marginal expression [90].

What are type 2 alveolar cells and what is their importance? As described by Robert J. Mason [91], at stage 3 of the COVID-19 disease, hypoxia and ground-glass infiltrates develop with a progression towards acute respiratory distress syndrome (ARDS). At this stage, the virus reaches the gas exchange units or alveolae and infects preferentially type 2 alveolar cells similarly to the influenza virus. Infected type 2 cells release further viruses, contribute to the infection of nearby alveolae, and later undergo apoptosis. The same author in a later article [92] explains relevant characteristics of type 2 alveolar cells. These cells are involved in keeping the alveolae fluid-free through their specific expression of CFTR chloride ion transporter. Type 1 alveolar cells contribute to maintaining the fluid-free state of the alveolae through CLIC5, another chloride ion transporter. These statements are confirmed by single-cell RNA seq data as presented here [90] and here [93].

Type 2 alveolar cells have various functions. In Figure 1 we present the distinctive role of the type 2 alveolar cells and their role in surfactant production and SARS-CoV-2 virus replication. They can be considered the stem cells of the alveolae [94] that can both regenerate themselves and also give rise to the type 1 alveolar cells, those flat cells that cover the largest part of the alveolae. Their overall differentiation takes up to one year. Another major function of these cells is that they are the ones producing the surfactant, a mixture of different lipids that cover the alveolae that is essential in keeping them open during respiration. The surfactant is a mixture of various lipids stored in the lamellar bodies of type 2 cells, organelles that can be considered as modified lysosomes. These lamellar bodies release their content by exocytosis. The released surfactant is in part degraded by macrophages or a smaller part is recycled by type 2 alveolar cells [95]. The critical connection between surfactant metabolism and lysosomal storage diseases was described in detail by Tamara Paget, Emma J. Parkinson-Lawrence, and Sandra Orgeig [96]. The apoptosis of the viral infected type 2 cells could affect the respiratory function in many ways. With the decrease in surfactant production, the alveolae could collapse, and as a result of the diminished chloride ion transport, fluids might accumulate in the alveolae, and by the destruction of the stem cell pool of the alveolae, the long-term regenerative potential of the lung might be diminished. As a conclusion, we can state that type 2 alveolar cells are critical in maintaining the physiological conditions of the lung, they are targets of the SARS-CoV-2 virus and express the transporter of fluoxetine, therefore they are excellent candidates for the protective effects of fluoxetine in case of COVID-19.

### 5.5. Mechanisms for Lysosome Enrichment of Fluoxetine

In addition to a likely high intensity of the transport of fluoxetine by OCT3/SLC22A3 into alveolar cells, other mechanisms might lead to high levels in the lung, such as (1) phospholipid binding and (2) lysosomal trapping [97]. The experiments that suggest these mechanisms were performed in different types of rat tissue slices immersed in media containing fluoxetine. From a medium containing 5 µM of fluoxetine, the accumulation ratio was highest in the lung tissue slices (75.6%) and moderate accumulation (10.5%) was observed on brain tissue slices. This accumulation was reduced by roughly a quarter by adding lysosomal inhibitors such as monensin or NH_4_Cl. Similar accumulation in the lung was seen for promazine, imipramine, amitriptyline, sertraline, and carbamazepine, too. In this early study, as a mechanism for the lung accumulation of the drug in lung tissues, it was suggested that the abundance of lysosomes in the lung alveolar macrophages, together with the abundance of surfactant rich in phospholipids, might be responsible. At the same time, although the brain does contain a high amount of phospholipids, it showed only moderate accumulation of these drugs. According to the early hypothesis by Korhuber [98], the slow accumulation of lysosomotropic psychoactive agents in the brain might contribute to the delayed effects observed in clinical practice, and may disturb several biochemical processes that require an acidic milieu, such as the proton-driven transport of monoamines into synaptic vesicles. As such, the accumulation of fluoxetine as a lysosomotropic agent in lung tissue might dramatically change the function of lysosomes, as described earlier in this review in Section 5.4.

The hypothesis that lysosomotropic agents could be beneficial in fighting COVID-19 was elaborated in detail in several papers, including one by Homolak in June 2020 [19] where some clinical studies involving lysosomotropic agents were also listed. The topic was elaborated in the same month by Blaess [99] as part of potential new therapeutic strategies, where a list of proposed lysosomotropic agents was presented. One of the proposed representatives of the lysosomotropic agents was Azithromycin. This antibiotic was tested in the early phases of the COVID-19 pandemic for preventing severe symptoms, but no statistically significant results could be shown [100]. Chloroquine and hydroxychloroquine, also lysosomotropic agents, were widely investigated for preventing severe outcomes of COVID-19. Currently, these drugs are not recommended for the treatment of COVID-19 in combination with Azithromycin, as a large body of evidence, such as the RECOVERY trial [77], the Solidarity trial [101], and the PETAL trial [102], proved their ineffectiveness in COVID-19. The NIH COVID-19 treatment guidelines as of early 2022 do not recommend these drugs to be used to prevent severe COVID-19 outcomes [103]. These observations are against the hypothesis that lysosomotropic agents, in general, could help prevent severe COVID-19 outcomes, and suggest that other mechanisms might be responsible for the potentially beneficial effects observed in the case of fluoxetine and fluvoxamine in the clinical studies presented.

## 6. Real-Life Clinical Aspects of Using Fluoxetine and Fluvoxamine for the Treatment of COVID-19

The possibility of using fluoxetine and fluvoxamine, well-known SSRI antidepressants, in the treatment of COVID-19 more extensively is a daunting prospect as potential difficulties and caveats have to be considered. Despite their favourable safety profile, these medications still have side effects, and little is known about their interaction with antiviral and other anti-COVID-19 medications [104,105]. Another important question arising during this phase of the pandemic is whether these adjuvant medications will be needed as SARS-CoV-2 becomes a better-controlled virus at a more endemic scale, rather than an uncontrollable pandemic. Regulations of off-label medication vary between countries and the COVID-19 situation resulted in even more divergent practices in different parts of the world. Therefore, it is important to consider the local regulations and practices for each country in this regard.

### 6.1. Risk Evaluation

As seen in the description of the clinical studies, most applied some degree of psychiatric oversight. SSRI antidepressants have been used in 10 to 100 M patients since their introduction and they have proven to be safe. They are not medications without side effects, however. [106]. Psychiatrists have widespread experience regarding these medications and have also developed screening tools and skills to predict when to limit their use in certain patients. Therefore, a psychiatric examination and safety assessment seems strongly recommended before the administration of these medications in COVID-19 patients, regardless of their psychiatric history.

### 6.2. Psychiatric Side Effects

Due to their potent anti-depressive effects, SSRIs can induce a manic switch or a mixed episode, or a long-term condition called rapid cycling in patients with an undiagnosed bipolar affective disorder [107]. Psychiatric evaluation should assess previous manic or hypomanic episodes, substance abuse, or positive family history. Moreover, SSRIs can induce akathisia or agitation, i.e., restlessness or increased psychomotor activity. In elderly patients this state can lead to confusion. Suicidal risk should be monitored throughout the administration of SSRIs as baseline suicidal ideation combined with agitation can lead to increased suicidal risk. Antidepressants improve mood and decrease the risk of suicide, but it is important to monitor the dynamics of suicidal symptoms, as these medications are normally used in a patient group at high risk of suicide, where even slight side-effects can prove intolerable. The Columbia Suicidality Rating Scale (CSRS) is a well-established instrument for screening patients regarding suicidal risk [108]. Psychiatric side effects are dose-dependent, therefore, a gradual titration of fluoxetine and fluvoxamine under psychiatric control is necessary.

### 6.3. Somatic Side Effects

Common somatic side effects include nausea and abdominal discomfort, and in rare cases, diarrhoea and vomiting. Headache, insomnia, drowsiness, and dry mouth can also occur. Urinary retention is also seen in very rare instances. Hyponatremia is developed in rare cases that will cease after discontinuation of SSRIs. Due to the effects of SSRIs on platelet functions, bleeding in SSRI-treated patients has been thoroughly investigated [109,110], demonstrating that there is a positive association between haemorrhagic complications and SSRI-administration, especially in anticoagulated, non-steroidal anti-inflammatory drug-taking patients or patients with hepatic cirrhosis, but these cases are extremely rare. Given the fact that the vast majority of COVID-19 patients receive low molecular weight heparin or novel per os anticoagulant treatment, haemorrhagic side effects have to be carefully monitored in the case of SSRI co-administration. Sexual side effects, such as anorgasmia, erectile dysfunction, decreased libido, and chronic side effects such as the increased risk of osteoporosis seem irrelevant in the context of treating COVID-19.

### 6.4. Drug Interactions

The administration of fluoxetine and fluvoxamine in the COVID-19 patient population raises several questions about drug interactions [111]. These medications as potential COVID-19 therapies have been identified based on their receptor affinity profile and by serendipity after their administration as antidepressant and anxiolytic agents. If they become part of the general COVID-19 treatment protocol, they will be used together with other medications that have previously not been co-administered with SSRI antidepressants. In the following paragraph, we summarize preliminary findings of drug-drug interactions.

The antiviral agent remdesivir, which is used worldwide for the treatment of COVID-19, is metabolized by both cytochrome enzymes CYP3A4, CYP2D6, and CYP2C8, and hydrolases, such as carboxylesterases [105]. Fluvoxamine is an inhibitor of CYP1A2 and CYP3A4, fluoxetine inhibits CYP2D6, therefore, both investigated SSRIs can lead to elevations in the plasma level of remdesivir. At the same time, remdesivir itself can inhibit CYP1A2. Liver enzymes and drug plasma levels should be monitored during the co-administration of these drugs. The effects of dexamethasone, another widely used medication in COVID-19 patients with pneumonia, can cause pharmacodynamic interactions with SSRIs because they can both cause confusion, psychosis, and agitation.

Other anti-COVID-19 agents, such as tocilizumab, baricitinib, and bamlavinab, use other pathways to exert their effect, therefore are not likely to interact with fluvoxamine or fluoxetine.

Currently on the horizon, Paxlovid is a promising SARS-CoV-2 inhibitor that targets the viral main protease (Mpro). The inhibitor is intended for oral administration, which is composed of the dipeptidyl inhibitor PF-07321332 (nirmatrelvir) in combination with a booster protease inhibitor (ritonavir). Additionally, the IV formulation of the drug is also undergoing clinical trials. Gaining emergency use authorization by the FDA in late December 2021, this drug is close to mass marketing and may well be the only effective antiviral in the context of treatment and prophylaxis against COVID-19 [112]. Nirmatrelvir is a substrate of CYP3A4 but when administered with ritonavir, a potent inhibitor of CYP3A, it undergoes minimal metabolism, enhancing its serum levels. It is indeed very early to predict how co-administration of SSRIs with Paxlovid may affect drug serum levels and whether or not cytotoxicity issues may arise; these, however, deserve a thorough examination.

## 7. Summary

We have provided an overview of the experimental, pharmacokinetic, and clinical evidence regarding the potential usage of fluoxetine and fluvoxamine in the treatment of COVID-19. Although treatment protocols vary greatly across countries, the presented findings are suggestive of a significant protective effect of these SSRIs in COVID-19 patients. We can hope that additional clinical studies will demonstrate conclusively whether the use of these antidepressants in COVID-19 is efficient and clinically meaningful. At this point, we can claim that both fluoxetine and fluvoxamine can provide a small but significant benefit for patients with COVID-19. Further studies are needed to show whether the benefits of receiving fluoxetine and fluvoxamine are more on the preventive side, i.e., the inhibition of the viraemic and inflammatory stage of the disorder, or they produce long-term effects, e.g., the prevention of pulmonary fibrosis.

Even if the efficacy of fluoxetine and fluvoxamine is accepted, a number of questions remain. For example, are other SSRI and non-SSRI antidepressants as effective as fluoxetine or fluvoxamine? If not, what is their unique pharmacological property that is exerting their efficacy? Is there a difference favouring either fluvoxamine or fluoxetine? Which COVID-19 patients should receive this type of treatment? Is there a special patient profile or should we administer fluoxetine and fluvoxamine for all patients? Given the side effects of these drugs, we can state that bipolar patients should be excluded and suicidality must be monitored. Elderly patients are an important group given their higher risk for severe COVID-19, however, the side effects can also be more severe in this group.

Another challenging aspect of the COVID-19 pandemic continues to be the fact that the SARS-CoV-2 virus is still undergoing mutations. We hope that subsequent mutations will lead to the domestication of the virus, but this is not necessarily the only scenario. We can speculate that fluoxetine and fluvoxamine could offer an important treatment option in patients undergoing mild COVID-19, which is typically caused by the most recent omicron virus variant. Overall, in our view, these common SSRI antidepressants represent an important opportunity for being better prepared for different outcomes of the COVID-19 pandemic.

## Figures and Tables

**Figure 1 ijms-23-03812-f001:**
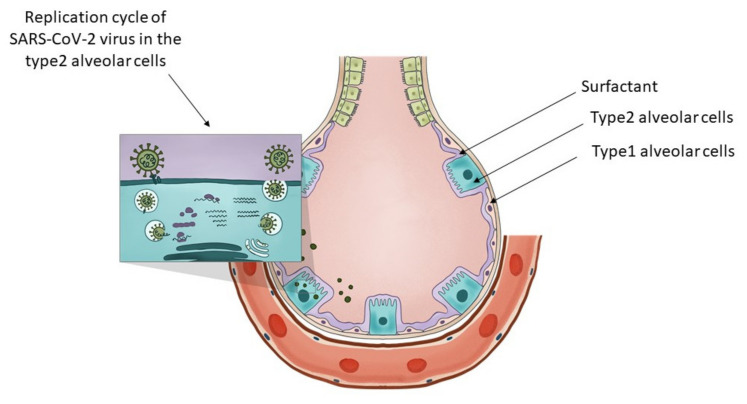
The key role of type 2 alveolar cells in the surfactant production and SARS-CoV-2 infection at the level of the lung alveolae. The insert on the top-left presents the major steps of the replication cycle of SARS-CoV-2 virus in the type 2 alveolar cells.

**Table 1 ijms-23-03812-t001:** Summary of the available clinical data on the potential benefit of fluoxetine and fluvoxamine in preventing severe COVID-19.

Study	Author	Study Design	Type of Antidepressant	Number of Patients Enrolled	Primary Endpoints	Results
Fluvoxamine vs. Placebo and Clinical Deterioration in Outpatients with Symptomatic COVID-19	Lenze et al. [49]	Randomized, double-blinded, placebo-controlled study	Fluvoxamine 3 × 100 mg (15 days)	N(FLUV) = 80, N(PLC) = 72	Clinical deterioration (dyspnoea or low saturation level)	0/80 (FLUV) vs. 6/72 (PLC)
Prospective Cohort of Fluvoxamine for Early Treatment of Coronavirus Disease 19	Seftel et al. [50]	Prospective, open-label, real-world, cohort study	Fluvoxamine 2 × 50 mg (14 days)	N(FLUV) = 65, N(Obs) = 48	Hospitalisation, ICU/death, symptoms on day 14	Hosp.: 0/65 (FLUV) vs. 6/48 (Obs) ICU: 0 vs. 2 Death: 0 vs. 1 Symp. on D14: 0/65 vs. 29/48
Effect of early treatment with fluvoxamine on risk of emergency care and hospitalisation among patients with COVID-19: the TOGETHER randomised, platform clinical trial	Reis et al. [51]	Randomized, double-blinded, placebo-controlled study	Fluvoxamine 2 × 100 mg (10 days)	N(FLUV) = 741, N(PLC) = 756	Primary: hospitalisation and emergency care setting (longer than 6 h)	Primary endpoint: 10.66% (FLUV) vs. 15.75% (PLC)
Fluoxetine use is associated with improved survival of patients with COVID-19 pneumonia: a retrospective case-control study	Németh et al. [52]	Retrospective, case-control study	Fluoxetine 1 × 20 mg	N(FLUO) = 110, N(TAU) = 159	Death	Mortality: 13.6% (FLUO) vs. 23.8% (TAU) *p* = 0.002
Association between antidepressant use and reduced risk of intubation or death in hospitalized patients with COVID-19: results from an observational study	Hoertel et al. [57]	Retrospective, cohort study	All antidepressants	N(AD) = 345, N(SSRI) = 195, N(non-SSRI) = 150, N(control) = 6885	Intubation/death	Antidepressant use was significantly associated with reduced risk of intubation or death.
Mortality Risk Among Patients With COVID-19 Prescribed Selective Serotonin Reuptake Inhibitor Antidepressants	Oskotsky et al. [58]	Retrospective, cohort study	All SSRIs	N(all) = 490373, N(SSRI) = 3401, N(FLUO + FLUV) = 481	Death	Mortality: 14.6% (SSRI) vs. 16.3% (control) 10% (FLUV+FLUO) vs. 13.3% (control) 15.4% (other SSRI) vs. 17% (control)

Abbreviations: FLUV: fluvoxamine, FLUO: fluoxetine, PLC: placebo, Obs.: observation, ICU: intensive care unit, Symp.: symptomatic, TAU: treatment as usual, AD: antidepressants, SSRI: selective serotonin reuptake inhibitor.

## Data Availability

Not applicable.

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
