# Peer review of "Potential Role of the Antidepressants Fluoxetine and Fluvoxamine in the Treatment of COVID-19"

_ijms, 2022, doi:10.3390/ijms23073812_

Round 1

Reviewer 1 Report

Dear authors, thank you for the interesting study. I think that the idea is great. I have only some minor suggestions.

  1. Probably you can shorten the introduction part of SARS biology and COVID-19 pathogenesis, as many reviews were published.
    1. Nile, S. H., Nile, A., Qiu, J., Li, L., Jia, X., & Kai, G. (2020). COVID-19: Pathogenesis, cytokine storm and therapeutic potential of interferons. Cytokine & growth factor reviews, 53, 66-70.
    2. Gusev E, Sarapultsev A, Solomatina L, Chereshnev V. SARS-CoV-2-Specific Immune Response and the Pathogenesis of COVID-19. Int J Mol Sci. 2022 Feb 2;23(3):1716. doi: 10.3390/ijms23031716.
    3. Mason, R. J. (2020). Pathogenesis of COVID-19 from a cell biology perspective. European Respiratory Journal, 55(4).

  1. In part 1.A Hungarian, retrospective, case-control study of Németh et al you described the stress-lowering effect and the benefits of antidepressant/stress-decreasing action of fluoxetine on the course of COVID-19. Possibly, you can describe the antidepressant effect as the distinct action of which fluoxetine and fluvoxamine might prevent the development of severe COVID-19. You can find a lot of literature linking the severity of depression with the course of diseases (not only viral infections). Moreover, there are several papers on the COVID and depression as well.
    1. Dąbrowska E, Galińska-Skok B, Waszkiewicz N. Depressive and Neurocognitive Disorders in the Context of the Inflammatory Background of COVID-19. Life (Basel). 2021 Oct 8;11(10):1056. doi: 10.3390/life11101056. PMID: 34685427; PMCID: PMC8541562.
    2. Perlmutter A. Immunological Interfaces: The COVID-19 Pandemic and Depression. Front Neurol. 2021 Apr 23;12:657004. doi: 10.3389/fneur.2021.657004. PMID: 33967944; PMCID: PMC8102701
  2. I can suggest discussing the studies of Nazimek and Mayes, who were one of the first have highlighted the direct impact of various antidepressant drugs on macrophages and displayed that the administration of fluoxetine, venlafaxine and moclobemide have been resulted in the suppression of humoral and cell-mediated immunity.
    1. Nazimek K, Kozlowski M, Bryniarski P, Strobel S, Bryk A, Myszka M, Tyszka A, Kuszmiersz P, Nowakowski J, Filipczak- Bryniarska I (2016): Repeatedly administered antidepressant drugs modulate humoral and cellular immune response in mice through action on macrophages. Exp. Biol. Med. 241, 1540–1550
    2. https://doi.org/10.1177/1535370216643769
    3. Nazimek K, Strobel S, Bryniarski P, Kozlowski M, Filipczak- Bryniarska I, Bryniarski K (2017): The role of macrophages in anti-inflammatory activity of antidepressant drugs. Immuno¬biology 222, 823–830
    4. https://doi.org/10.1016/j.imbio.2016.07.001
    5. Maes M (2011): Depression is an inflammatory disease, but cell-me¬diated immune activation is the key component of depression. Prog. Neuropsychopharmacol. Biol. Psychiatry 35, 664–675
    6. https://doi.org/10.1016/j.pnpbp.2010.06.014

  1. Costa LHA, Santos BM, Branco LGS. Can selective serotonin reuptake inhibitors have a neuroprotective effect during COVID-19? Eur J Pharmacol. 2020 Dec 15;889:173629. doi: 10.1016/j.ejphar.2020.173629. Epub 2020 Oct 3. PMID: 33022271; PMCID: PMC7832208.

Reviewer 2 Report

Please provide the bibliography for this paragraph „Severe COVID-19 develops in several steps and might require hospitalization or in 
many cases intensive care. It is of major importance to identify ways that can prevent the development of severe COVID-19 and decrease the burden on the healthcare system in general and intensive care treatment facilities in particular. Several potential medications have been tested during the pandemic period based on prior knowledge about the life 
cycle of the SARS-CoV virus. Drugs known to be lysosomotropic agents have been tested, such as azithromycin, chloroquine, hydroxychloroquine, and ivermectin, but were found to be ineffective in preventing severe COVID-Other drug repurposing efforts were highly successful; such as using steroids to prevent the cytokine storm, budesonide to reduce local inflammation in the lung, and anti-IL-6 inhibitors, such as tocilizumab to prevent the 
devastating results of an already initiated cytokine storm.„

The introduction is not updated to the current period.

Pkease provide bibliography for this paragraph „Modulation of endolysosomal pH could potentially impair the formation of viral replication complexes, in addition to impeding viral trafficking and budding. Both fluoxetine and fluvoxamine; among other SSRIs, are considered to be lysosomotropic agents, and hence, it is plausible 
that lysosomotropic and endolysosomal pH-modulating effects of these molecules may indeed show potential beneficial effects in the context of COVID-19. „

The authors do not specify the figure number ”Figure The key role of type 2 alveolar cells in the surfactant production and SARS-CoV-2 infection 
at the level of the lung alveolae. The insert on the top-left presents the major steps of the replication cycle of SARS-CoV-2 virus in the type 2 alveolar cells.„ and also they do not specify it in the text.

The table aso has no number „Table Summary of the available clinical data on the potential benefit of fluoxetine and fluvoxamine 
in preventing severe COVID-19. „ and is not mentioned in the text.

I would also suggest the authors to restructure a little bit the text to be more clear.  
